# Which are the Nutritional Supplements Used by Beach-Volleyball Athletes? A Cross-Sectional Study at the Italian National Championship

**DOI:** 10.3390/sports8030031

**Published:** 2020-03-11

**Authors:** Stefano Amatori, Davide Sisti, Fabrizio Perroni, Samuel Impey, Michela Lantignotti, Marco Gervasi, Sabrina Donati Zeppa, Marco B. L. Rocchi

**Affiliations:** 1Department of Biomolecular Sciences, Service of Biostatistics, University of Urbino Carlo Bo, Piazza Rinascimento 7, 61029 Urbino, Italy; s.amatori1@campus.uniurb.it (S.A.); davide.sisti@uniurb.it (D.S.); m.lantignotti@campus.uniurb.it (M.L.); marco.rocchi@uniurb.it (M.B.L.R.); 2Department of Biomolecular Sciences, Section of Exercise and Health Sciences, University of Urbino Carlo Bo, Via I Maggetti 26/2, 61029 Urbino, Italy; marco.gervasi@uniurb.it (M.G.); sabrina.zeppa@uniurb.it (S.D.Z.); 3School of Medical and Health Sciences, Building 21, Edith Cowan University, 270 Joondalup Drive, Joondalup, WA 6027, Australia; sam.impey95@gmail.com

**Keywords:** athletic performance, dietary supplements, sports nutrition sciences, volleyball

## Abstract

Beach volleyball is an intermittent team sport played under high temperature and humidity. Given that some nutritional supplements can enhance sports performance, this study aimed to evaluate the quantity and the heterogeneity of the nutritional supplementation practices of amateur (n = 69) and professional (n = 19) beach volley athletes competing in the Italian National Championship; an online form was used to collect data about the supplementation habits. The latent class analysis was used to find sub-groups characterised by different habits regarding supplements consumption. The most frequently used supplements (more than once a week) are vitamins B and C (39.2% of athletes), protein (46.8%), and caffeine (36.9%). The latent class analysis revealed three different sub-groups of athletes: the first class (56.7%) included athletes who were used to take very few supplements, the second class (17.0%) was characterised by higher consumption of supplements and the third class (26.2%) was in the middle between the others two. Groups were characterised not only by the quantity but also by the category of supplements used. Our results highlighted a high heterogeneity in supplementation habits. A pragmatic approach to supplements and sports foods is needed in the face of the evidence that some products can usefully contribute to enhancing performance.

## 1. Introduction

Athletes have to train as hard as possible with optimal adaptation and recovery, to remain healthy and injury-free, to achieve a physique that is suited to their event, and to perform at their best on the day(s) of peak competitions [1]. Beach volleyball is an intermittent team sport played by two teams of two players on a sand court divided by a net [2]. It is characterised by frequent high-intensity efforts interposed by short recovery phases [3]. The performance involves jumps (e.g., attacking, serving, blocking), short sprints, changes of direction and diving digs [4]. During a single set, Palao et al. [3] observed that defenders and blockers performed an average of 27 and 31 jumps, respectively. In addition, moving on sand increases energy cost compared to moving on the solid ground [5]. Beach volleyball is played under demanding environmental conditions: Zetou et al. [6] reported that during over 50 matches analysed in an official tournament, the mean air temperature was 33.6 °C (max 38 °C) and mean humidity was 56% (max 75%). Bahr and Reeser [7] reported that in the 19% of the World Tour tournaments between 2009 and 2011, wet-bulb globe temperature (that combine temperature, humidity, wind speed and solar radiation) exceeded 32 °C, that is considered the upper limit for physical activity in hot and humid conditions (US Navy ‘black flag’ condition). Players have to play up to three matches (42.2 ± 9.8 min per match; ranging from 30 to 64 min) per day with small intervals between games and for three days in a row [8]. The extended duration of exposure of the players to the sun and to high temperatures are important factors that increase the risks of dehydration and thermal stress. 

In conjunction with training and nutrition, numerous evidences demonstrate that the appropriate ingestion of some nutritional supplements can enhance sports performance [9]. A number of excellent reviews have evaluated the performance-enhancing effects of most supplements in endurance sports [9,10,11]. On the contrary, less attention has been paid to the performance-enhancing claims of supplements in the context of team-sport performance; only a few papers in the literature summarised the effects of sports supplements in this field [12,13]. Even if some generic evidence is present for carbohydrate [14] and protein [15] supplementation in team sports, supplements that enhance performance in some disciplines may not work in the same way in others (and vice versa). For example, nutritional supplements that have been demonstrated to improve continuous exercise performance may not improve intermittent exercise performance. According to Maughan et al. [9] nutritional supplements can be classified as dietary supplements, sport nutrition products and ergogenic supplements. The category of dietary supplements mainly comprises micronutrient supplements such as vitamins and minerals, but also essential fatty acids. Such dietary supplements may promote the athletes’ general health through the prevention and treatment of nutrient deficiencies [11]. Sport nutrition products mainly contain macronutrients, such as carbohydrates, proteins and fat, and include sports drinks, recovery drinks, energy bars, etc. The category of ergogenic supplements mainly comprises products with performance-enhancing claims, such as caffeine and creatine [16]. Furthermore, some supplements may be used for multiple functions. For example, carbohydrate supplements are used to enhance performance in many events via the provision of fuel substrate [17] or to support the immune system [18].

Proposed factors responsible for performance decrements during high-intensity periods of play during team-sports include, among others, limitations to energy supply and metabolite accumulation, that limit performance by causing fatigue, loss of concentration and consequent reduction of skills over the course of the event [10,19]. Supplements that offset the influence of these limiting factors are able to improve the recovery of both sprint and jump performance and to speed the recovery of athletes following matches or training, subsequently improving team-sport performance [12]. In fact, performance enhancement, prevention of nutritional deficiencies, better body composition, immune system enhancement, and recovery from training and injury are some of the known reasons why athletes use supplements [9]. 

To our knowledge, no studies have been conducted investigating the use of nutritional supplements by beach volley athletes. Thus, the purpose of this study was to evaluate the quantity and the heterogeneity of the nutritional supplementation practices of amateur and high-performance beach volley athletes participating in the Italian National Championship. Aspects of dietary supplementation that were evaluated included supplement use, reasons for supplementation, and sources of supplementation information.

## 2. Methods

This was an observational cross-sectional study involving amateur and professional beach volley athletes competing in the Italian National Championship in the 2018 season. After a verbal and written explanation of the experimental design of the study, approved by the local Institutional Review Board, written informed consent was obtained from the athletes, and a trained research assistant sent an online form (Google Form) by email to every athlete. Participants responses were anonymous, as no identifiable details were collected online. In addition, all players were fully accustomed to the procedures used and were informed that they could withdraw from the study at any time. 

### 2.1. Participants

At the final of the Italian National Championship (Catania, 2018), all the 134 athletes participating at the competition were recruited for participation in the study. The response rate was 86.6%, as 116 athletes that were approached took part in the study. Of these, 28 athletes (20.9%) were excluded because they did not answer correctly to the questions. 88 athletes (65.7%) were then included in the final analyses. Of the 88 athletes, 40 (45.5%) were males, and 48 (54.5%) were females. Height was 190.4 ± 6.5 cm for males, 174.3 ± 5.4 cm for females; weight was 84.8 ± 7.3 kg for males, 63.0 ± 5.4 kg for females. The mean age of the athletes was 28.1 ± 6.0 (min 18; max 47) years. Only 3 subjects (age = 41; 46; 47 years) were over the age of 40. Additional demographic characteristics of the subjects were collected, education level, occupation, years of sports practice, level of competition and sponsorship by a sports supplement producing company. Participants’ characteristics are shown in Table 1.

### 2.2. Procedures and Instruments

The procedure for data collection that was used in the current study has been used before [20]. The online form was divided into two main sections. The first section captured information on participants’ socio-demographic and sports history-related characteristics. The second section consisted of questions about athlete nutritional supplement use, including the type of supplements used, frequency of consumption, personal motivations for use, and the sources of sports nutrition information. Subjects were asked to indicate the frequency of use on a 3-points Likert Scale (0 = <1 time/week, 1 = 1 to 2 times/week; 2 = >2 times/week), of each of the following supplement: carbohydrate, protein, branched-chain amino acids (BCAA), caffeine, glutamine, creatine, carnitine, sodium bicarbonate, beta-alanine, probiotics, phosphates, vitamins (B, C, D, E), calcium, iron, omega-3, omega-6 and herbals. At the end of the list, the variable “Other” was present to give the possibility to the subject to add any other supplement. Moreover, subjects were asked to answer “yes” or “no” if they were used to use each of these products during the match: sports drinks, electrolytes, energy drinks, sports bar and sports gel. Finally, subjects were asked to select one (or more) option from a list, about where they search/take information about the nutritional supplements and their personal motivation to their use.

### 2.3. Statistical Analysis

SPSS Statistics 22.0 was used for analysis, and a *p* < 0.05 was considered statistically significant. Continuous variables, such as age and years of sports practice, were categorised in classes and the data are presented as absolute frequencies and percentages. Latent class analysis (LCA) was performed with R 3.5.3 and poLCA package version 1.4.1 [21]. It has been used to derive a profile of beach volleyball athlete from the supplement items above reported. LCA identifies (latent) subgroups of individuals (i.e., classes) who share common features, whereas factor analysis identifies unobserved common dimensions accounting for the correlations among observed variables. LCA posits that a heterogeneous group can be reduced to several homogeneous subgroups by evaluating and then minimising the associations among responses across multiple variables, and tests for the existence of discrete groups with a similar symptom or item endorsement profile [22]. LCA estimates two parameters: (1) the likelihood of endorsement of a given item for individuals in a particular class; and, (2) the class membership probabilities. Since in LCA no a priori assumptions are made concerning the number of latent classes, LCA model selection was conducted according to fitting indices such as G2 likelihood ratio (−2 × ln(L)), the Akaike information criterion (AIC), the Bayesian information criterion (BIC), and the sample-size adjusted BIC (SSABIC) [23]. For each of these indexes, lower values indicate better fit. Chi-square test was used to assess associations among supplements used and socio-demographic characteristics (gender, age, level of education, occupation, years of beach volley practice, level of competition and sponsorship); Cramer’s V was calculated as a measure of association.

## 3. Results

Complete results of the dietary supplement, sports nutrition products and ergogenic aids categories are reported in Table 2. None of the athletes reported using any supplements other than those already listed a priori in the survey.

### 3.1. Latent Class Analysis

The three-classes solution has been chosen on the basis of fitting indexes; AIC, SSABIC and G2 likelihood-ratio were lower in the three-classes solution than in the two-classes one, whilst BIC was higher in the four-classes solution with respect to the three-classes one (Figure 1).

In this solution, a first class (LC_1_) has been found, which included 50 athletes (56.7%) characterised by taking very few supplements; the participants included in LC_1_ had a 96.6% probability of taking supplements less than once per week. The second class (LC_2_, including 15 athletes, 17.0%) were characterised by higher consumption of supplements; the probability of the participants taking supplements one or two times per week was 15.0%, and 41.3% more than two times per week. The third class (LC_3_, including 23 athletes, 26.2%) was characterised by a higher probability of use supplements with respect to LC_1_, but lesser than LC_2_. In fact, it can be highlighted that the differences in supplements intake habits between group 2 and 3 were found in particular regarding to use of dietary supplements (all vitamins, calcium, iron and herbals) and of some ergogenic aids (caffeine, carnitine, phosphates and sodium bicarbonate), with LC_2_ subjects were more likely to use these supplements frequently. Although the difference was not significant between LC_2_ and LC_3_, a trend can be noticed in higher use of carbohydrate, protein, BCAA and Omega-6 by the LC_2_ group. Complete results are reported in Table 3.

### 3.2. Athletes Characteristics and Nutritional Supplement Use

Use of protein (χ^2^ = 10.282, *p* = 0.036, V = 0.365) and BCAA (χ^2^ = 15.527, *p* = 0.004, V = 0.441) supplements was significantly greater in males than females. In fact, more than 35% of women said they did not use BCAA, and 25% did not even take proteins. For all the other predictive variables used (age, level of education, occupation, years of beach volley practice, level of competition and sponsorship) no significance was found for any of the supplements used (p > 0.05).

### 3.3. Supplement Intake during a Competitive Beach Volleyball Match

The results of the answer to the question “Do you normally use these supplements during the game?" showed that almost half of athletes (45.5%) used sports drinks (in the form of powder or ready to drink liquid), electrolytes (as powder sachets or tablets) (37.2%), and energy drinks (ready-to-drink liquid or concentrated shot) (36.7%) during the match. Athletes reported using carbohydrates in the form of bars and gels, 22.4% and 20.3% respectively.

### 3.4. Sources of Information and Athletes’ Motivation for Supplement Use

Nutritionists/dietitians were the main sources of information for the use of supplements (31.6%), followed by internet and teammates (15.8% for both), physicians (12.3%), pharmacists (12.3%), coaches (7.0%) and only a few athletes claimed to have taken information from specialist magazines (5.3%). Most athletes used supplements with the aim of improving performance (32.5%), preventing nutritional deficiencies (25.0%) and improving recovery (20.0%). As a minor percentage, they were used as a supplement to the diet (7.5%), to improve health, reduce stress, improve the immune system, improve self-perception and increase muscle mass (2.5% for all the items). Relative frequencies (%) of these items are graphically reported in Figure 2 and Figure 3. 

## 4. Discussion

To our knowledge, this is the first study that aimed to analyse the quantity and the heterogeneity of the nutritional supplementation practices of amateur and high-performance beach volley athletes. The main result of our study was that three main sub-groups were found using an LCA, each group defined by a different behaviour regarding supplements consumption (both type and frequency of assumption); these results highlight a high heterogeneity in the consumption of the supplements among beach-volley athletes. 

Training, recovery and nutrition are the essential components of the sport performance. Once these basic factors are accounted for, the use of some evidence-based nutritional supplements may help athletes to improve performance [24]. Many athletes have a "win at all costs" mentality and will choose to use a supplement regardless of the possible side effects. Furthermore, many of the effects reported for some supplements are anecdotal and have not been supported by scientific evidence (e.g., carnitine or glutamine) [11,25]. The “more is better” philosophy when applied to supplements consumption, may lead to adverse effects, often due to inappropriate patterns of use by athletes (i.e., mixing and matching different products without regard to total doses nor possible negative interactions between ingredients) or by the lack of safety of the products. In addition, common supplements have been found to contain undeclared prohibited substances, that may cause a positive doping outcome [9].

Only a few studies investigated the supplements consumption habits of volleyball/beach volleyball athletes. Zapolska and colleagues [26] reported the nutritional supplements consumption habits of a sample of 17 professional volleyball athletes. The use of sports supplements was declared by 89% of the respondents. The most popular supplements were: protein (71%), carbohydrate (24%), vitamins and minerals (82%), amino acids (76%) and stimulants (47%), including coffee (65%) and caffeine tablets (41%); also BCAAs, creatine and glutamine were consumed. Zetou et al. [27] explored the common practices of 47 beach volleyball players (both elite and non-elite) regarding fluid, supplements and nutrition intake during a tournament. Interestingly, they reported that only very few players (about 25% of their participants) took supplements or other aids, most of them did not follow a systematic diet before their games and, most surprisingly, they were not aware of the association between dehydration and performance (and they did not have any plan concerning the intake of fluids during their games).

The findings described from Zetou and colleagues are consistent with our data as 45.5% of subjects normally use sports drinks, 37.2% take electrolytes, and 36.7% of energy drinks during their matches. Our results showed that a large part of the interviewed athletes (56.7%) do not take any type of supplement, as reported from the LCA. However, LCA also showed a group of athletes (LC_2_; 17.0%) that reported taking—more or less frequently—almost every supplement included in the study, despite few of the supplements demonstrating an evidence-based efficacy in improving sports performance (e.g., sports drinks, carbohydrate, creatine, caffeine, β-alanine, etc.) [11]. We did not report any difference related to the athlete’s status (professional vs. recreational); however, this result may not be reliable, due to the very low number of elite athletes included in our sample. Our results are in accordance with those reported by Zetou et al. [27], who did not find any difference in nutrition practices or fluid intake between elite and non-elite athletes, but in contrast with Wardenaar et al. [28], who found some differences in supplements use between elite and non-elite athletes of several disciplines.

Even when there is a robust literature on sports supplementation, it may not cover all applications that are specific to an event, environment, or individual athlete [10]. According to the ISSN Exercise and Sports Nutrition Review Update [11], there is a category of supplement with strong evidence to support efficacy, mainly composed by sports food (sports drink, bars and gels, electrolytes, …), and performance-enhancing supplements (caffeine, beta-alanine, creatine, sodium bicarbonate and phosphate). Despite generic evidence supporting their use, it is suggested that these supplements would be best used with an individualised and event-specific protocol. If limited to the world of team sports, such as beach volleyball, the list of supplements that have a scientifically proven effect on athletes’ performance narrows further. After a literature search on the principal databases (PubMed and Sport Discus), no studies have been found that investigated the efficacy of specific supplements in beach volley. Lamontagne-Lacasse et al. [29] studied the effect of creatine supplementation on jumping performance in elite volleyball players, concluding that creatine could increase the height in repeated jumps, attenuating the magnitude of muscular fatigue and offering a potential advantage during a match. Mielgo-Ayuso et al. [30] reported that oral iron supplementation prevented iron loss and enhanced strength in female volleyball players during the competitive season. Pfeifer et al. [31] did not find any difference in performance after a low-dose caffeine and carbohydrate supplement intake during a volleyball competition, even if, as stated by the authors, the dose of supplement (54 g carbohydrate plus 100 mg caffeine, divided into two separate doses) that was used may not have been sufficient to produce a positive effect.

In regard to dietary supplements consumption (mainly vitamins and minerals), research demonstrated that if an athlete is deficient, exercise capacity may be reduced; despite this, performance improvements due to vitamins/minerals supplementation when athlete’s status is adequate have not been reported [11]. Although a direct link between vitamins/minerals supplementation and performance is lacking, several nutrients (such as vitamin C and D) have been suggested to help athletes stay healthy, in particular during intense training periods. The athletes involved in heavy training period may have higher Recommended Dietary Allowance (RDA) of some vitamins; obtaining adequate vitamins through use of supplement appears to be a prudent behaviour for some athletes [32].

### Source of Information and Motivation to Use

Athletes’ supplementation practices are often guided by family, friends, teammates, coaches, the Internet, and retailers, rather than sports dietitians and other sport science professionals [10]. A good percentage of the athletes reported to take information from a professional figure (nutritionist or *physician*), but the remaining 56.1% did not. Specifically, our results showed that 31.6% of the interviewed athletes reported having received a consultation by a nutritionist/dietitian and 12.3% by a physician; 15.8% reported asking information about supplements from internet and teammates, and 7.0% from the coach. Following advice from coaches, teammates, internet or non-scientific magazines may lead to bad supplement choices [33] and could present significant potential risks for the athlete health and performance. Our results are in accordance with Braun et al. [34] which showed that the motivations for supplement use include enhancement of performance or recovery, improvement or maintenance of health, an increase in energy, compensation for poor nutrition, immune support, and manipulation of body composition. In our study, 32.5% reported using nutritional supplements to improve performance, to prevent deficiencies (25.0%) and to enhance recovery (20.0%), as reflected by the most frequently used supplements (vitamins, proteins, BCAAs, carbohydrates, caffeine, glutamine and creatine). Nevertheless, Petróczi et al. [35] reported that, despite the frequent use of dietary supplements, athletes have misconceptions about their effectiveness; indeed, the authors did not observe agreement between athletes’ rationale and behaviour in relation to their nutritional supplement use. According to some other studies, there is a large population of athletes who report incorrect information about the supplements they use [36]. It is important that coaches, nutritionist, and athletes above all, make reference to the more recent sports nutrition and supplementation guidelines, published by the American College of Sports Medicine [10], the International Society of Sports Nutrition [11], and the International Olympic Committee [9].

The conditions in which beach volley matches are played make it a unique discipline that needs to be further studied. In particular, the environmental conditions (very hot and high humidity) highlight the need for proper integration and maintenance of fluid replacement to compete at best. The development of hyperthermia during exercise in hot ambient conditions is associated with a rise in sweat rate, which can lead to progressive dehydration if fluid losses are not minimised by increasing fluid consumption. Dehydration of even 2% body mass might impair sports performance [37,38]. Fluid replacement recommendations should be adapted at the particular environmental situation (temperature and humidity) of each match-day.

Understandably, this study was subject to some limitations. The sample size was quite low and very variable results were likely, with consequences difficult to make a reliable synthesis. Most athletes use supplements because they believe their diet is not satisfactory; in this study, we did not study the content of athletes’ diets and for this reason, we were not able to make observations on the relationship between the use of supplements and the adequacy of the nutrients of the diet of each individual. For these reasons, further large-scale studies are recommended to ascertain the quality and the heterogeneity of the nutritional supplementation practices of beach volley athletes.

## 5. Conclusions

Team sports share the common feature of intermittent, high-intensity efforts, but experience marked variability of game characteristics between sports, and positions/playing styles within the same sport. This creates a diversity of physiological challenges and nutritional needs for team sport athletes [39]. According to the aim of this study, to explore the common supplementation practices of beach volley athletes, our results highlighted a high heterogeneity in the supplementation habits: most of them did not use supplements in a structured way, using them occasionally and often without a scientific rationale behind it. It is appreciable that most athletes rely on professional figures (nutritionists and/or physicians) for their nutrition and supplementations protocols; an even greater reliance on these professional figures could increase the intake of some evidence-based supplements and limit their use of other (without proven scientific efficacy), thus improving the effects on their performance and health status. A pragmatic approach to supplements and sports foods is needed in the face of the evidence that some products can usefully contribute to a sports nutrition plan and/or directly enhance performance.

## Figures and Tables

**Figure 1 sports-08-00031-f001:**
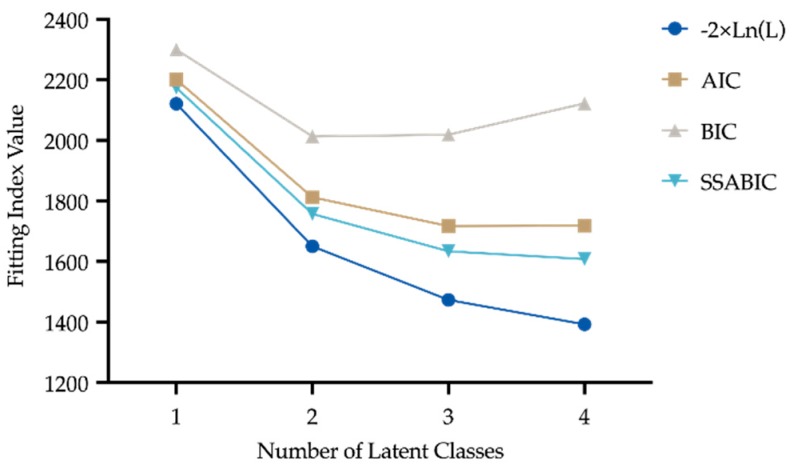
Latent Class Analysis scores (lower values indicate a better fitting). −2 × ln(L): G2 Likelihood ratio; AIC: Akaike information criterion; BIC: Bayesian information criterion; SSABIC: Sample-size adjusted BIC.

**Figure 2 sports-08-00031-f002:**
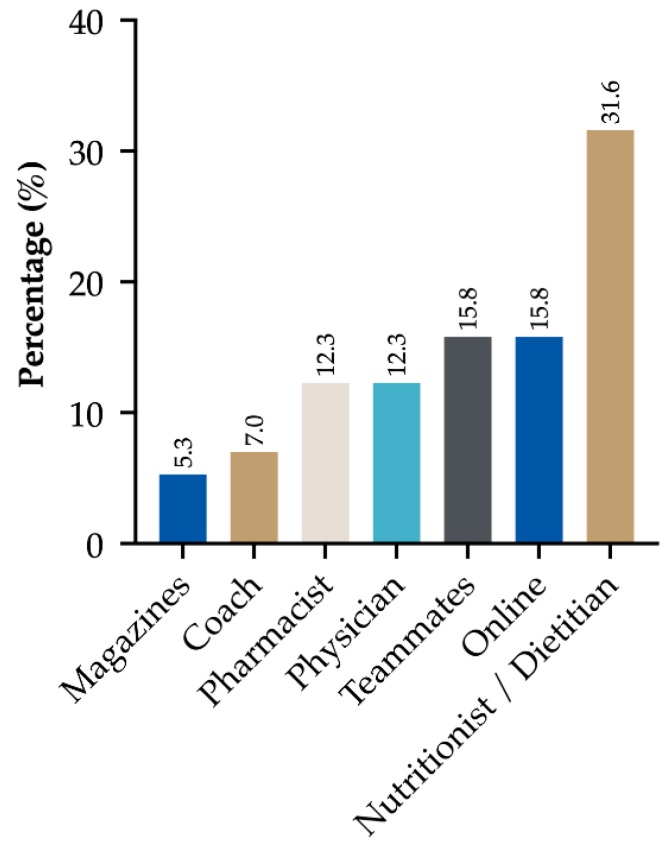
Athletes’ sources of information regarding supplements.

**Figure 3 sports-08-00031-f003:**
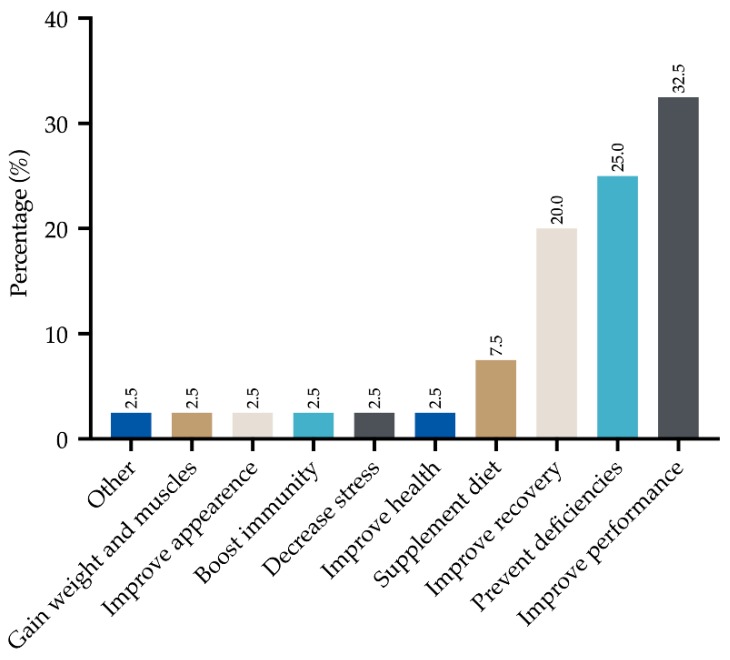
Athletes’ motivations to use supplements.

**Table 1 sports-08-00031-t001:** Socio-demographic characteristics of respondents (n = 88).

Variable		Frequency (n)	Percentage (%)
**Sex**	Males	40	45.5%
	Females	48	54.5%
**Age**	18–20 years	7	8.0%
	21–25 years	26	29.5%
	26–30 years	28	31.8%
	31–35 years	17	19.3%
	>35 years	9	10.2%
**Education Level**	High School	24	27.3%
	Bachelor Degree	34	38.6%
	Master Degree	30	34.1%
**Occupation**	Athlete	19	21.6%
	Independent Worker	23	26.1%
	Dependent Worker	28	31.9%
	Student	18	20.4%
**Duration Playing Beach Volley**	<5 years	29	33.0%
	5–10 years	45	51.1%
	>10 years	14	15.9%
**Level of Competition**	National	80	90.9%
	International	8	9.1%
**Sponsored**	Yes	8	9.1%
	No	80	90.9%

**Table 2 sports-08-00031-t002:** Type and frequency of use of nutritional supplements.

Dietary and Sports Supplements	<1 Time/Week	1–2 Times/Week	≥3 Times/Week
**Dietary Supplements**			
Vitamin C	60.8	16.5	22.8
Vitamin B	60.8	20.3	19.0
Vitamin D	67.9	16.7	15.4
Vitamin E	67.5	15.0	17.5
Calcium	70.1	14.3	15.6
Iron	72.7	14.3	13.0
Omega-3 Fatty Acids	74.0	15.6	10.4
Omega-6 Fatty Acids	87.5	9.7	2.8
Herbals	95.9	2.7	1.4
**Sports Nutrition Products**			
Protein	53.2	23.4	23.4
BCAA	62.5	18.8	18.8
Carbohydrate	66.2	18.2	15.6
**Ergogenic Aids**			
Caffeine	63.2	22.4	14.5
Glutamine	82.9	11.8	5.3
Creatine	81.6	9.2	9.2
Carnitine	91.9	5.4	2.7
Sodium Bicarbonate	94.4	4.2	1.4
Beta-Alanine	89.0	6.8	4.1
Probiotics	91.9	5.4	2.7
Phosphates	93.2	5.5	1.4

BCAA = Branched Chain Amino Acids.

**Table 3 sports-08-00031-t003:** Latent Class Analysis results: values represent the probability of response for members for each class.

**Class Sizes**		56.7%	17.0%	26.2%
		**LC_1_**	**LC_2_**	**LC_3_**
**Dietary Supplements**				
Vitamin B	1	**96.0%**	0.0%	39.2%
	2	2.0%	20.0%	**52.1%**
	3	2.0%	**80.0%**	8.7%
Vitamin C	1	**96.0%**	0.0%	39.4%
	2	2.0%	0.0%	**52.0%**
	3	2.0%	**100.0%**	8.7%
Vitamin D	1	**100.0%**	6.7%	52.3%
	2	0.0%	13.3%	**47.7%**
	3	0.0%	**80.0%**	0.0%
Vitamin E	1	**100.0%**	6.7%	48.0%
	2	0.0%	0.0%	**52.0%**
	3	0.0%	**93.3%**	0.0%
Calcium	1	**100.0%**	20.0%	52.3%
	2	0.0%	6.7%	**43.3%**
	3	0.0%	**73.3%**	4.3%
Iron	1	**100.0%**	20.0%	61.0%
	2	0.0%	13.3%	**39.0%**
	3	0.0%	**66.7%**	0.0%
Omega-3	1	**98.0%**	40.0%	56.7%
	2	2.0%	**26.7%**	**30.3%**
	3	0.0%	**33.3%**	**13.0%**
Omega-6	1	100.0%	60.0%	87.0%
	2	0.0%	**26.7%**	**13.0%**
	3	0.0%	**13.3%**	0.0%
Herbals	1	100.0%	86.7%	95.7%
	2	0.0%	**6.7%**	**4.3%**
	3	0.0%	**6.7%**	0.0%
**Sports Nutrition Products**				
Carbohydrate	1	**98.0%**	26.7%	39.3%
	2	2.0%	**20.0%**	**43.4%**
	3	0.0%	**53.3%**	**17.3%**
Protein	1	**84.1%**	13.3%	34.7%
	2	9.9%	**33.3%**	**34.9%**
	3	6.0%	**53.3%**	**30.4%**
BCAA	1	**90.0%**	40.0%	30.6%
	2	6.0%	13.3%	**43.3%**
	3	4.0%	**46.7%**	**26.1%**
**Ergogenic Aids**				
Beta-Alanine	1	98.0%	73.3%	87.0%
	2	2.0%	**13.3%**	**8.7%**
	3	0.0%	**13.3%**	**4.3%**
Glutamine	1	94.0%	66.7%	78.3%
	2	6.0%	**20.0%**	**13.0%**
	3	0.0%	**13.3%**	**8.7%**
Creatine	1	100.0%	46.7%	74.0%
	2	0.0%	**26.7%**	**13.0%**
	3	0.0%	**26.7%**	**13.0%**
Caffeine	1	**84.0%**	33.3%	56.6%
	2	8.0%	20.0%	**43.4%**
	3	8.0%	**46.7%**	0.0%
Carnitine	1	100.0%	73.3%	91.3%
	2	0.0%	**13.3%**	**8.7%**
	3	0.0%	**13.3%**	0.0%
Phosphates	1	100.0%	86.7%	87.0%
	2	0.0%	**6.7%**	**13.0%**
	3	0.0%	**6.7%**	0.0%
Sodium Bicarbonate	1	100.0%	80.0%	95.7%
	2	0.0%	**13.3%**	**4.3%**
	3	0.0%	**6.7%**	0.0%
Probiotics	1	94.0%	93.3%	91.3%
	2	2.0%	**6.7%**	**8.7%**
	3	**4.0%**	0.0%	0.0%

1 = <1 time/week; 2 = 1–2 times/week; 3 = >2 times/week; LC = Latent Class; BCAA = Branched-Chain Amino Acids. Values that are substantially higher for the segment than for the total sample are highlighted in bold. Values that are substantially lower for the segment than for the total sample are highlighted in italics.

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
