# Peer review of "Which are the Nutritional Supplements Used by Beach-Volleyball Athletes? A Cross-Sectional Study at the Italian National Championship"

_sports, 2020, doi:10.3390/sports8030031_

Round 1

Reviewer 1 Report

The subject of the studies was to assess the quantity and the heterogeneity of the nutritional supplementation practices in professional and non-professional beach volleyball players. Although the subject of the study is interesting and has practical application for sports nutritionists, there is some information that needs to be clarified.

Questionnaire:

You wrote in line 101 that the questionnaire was used before. In the study You referenced for ([17] in Your manuscript) authors declared that they asked about personal beliefs about supplements they used. Did You ask about that? If yes, why don't You discuss this part?

Did the participants have the opportunity to declare what supplement they are taking, or did You point some of the supplements in questionnaire and study participants were only declaring if they are consuming them or not? You achieved pretty good results in terms of using supplements with scientific proofed beneficial effects.

Study group:

Your group has a large age range. You declared that group age was 18-47 and in table I You pointed out that >35 was 9 participants. How many participants exactly were over 40? You need to reconsider if such a wide range does not interfere with resluts analysis (in case of the aim of supplementation - older people can have different purpose for diet supplementation than younger ones)

Discussion:

In lines 222-230 You discuss evidence for supplement use in volleyball players. First of all, it should be as separate paragraph since You discussed how supplements consumption looks like in other studies before. Second, You discussed only a few supplements Your study group was consuming. You should expand Your discussion section - please consider comparing supplements consumed by Your group with some recommendations. Perhaps the Australian Institute of Sport classification can be useful. Also, You can use your national recommendation (if You have those kinds of classifications/recommendations). Then You can discuss the justification of using these supplements in volleyball. Of course, not all supplements were tested in the case of volleyball and none of them in beach volley. But You can try to find studies in other sports with similar physical demands.

lines 237-240: You can't really say, that doctors, pharmacists or trainers do not necessarily have specific knowledge about supplementation if You not referenced it. You wrote that "they may not have specific knowledge (...) in the context of beach volleyball players" -dietitians also may not have that knowledge if they did not work with such a specific group. So You can't say that one group is bad because they may not have knowledge when You say the other is fine, when they actually may also have not this kind of practice. There is some evidence that after trainers advice there were some bad supplement choices (i.e. Dobrowolski & WĹ‚odarek: Assessment of the dietary supplement use by professional female football players. Polish Journal of Sports Medicine / Medycyna Sportowa . 2019, Vol. 35 Issue 1, 51-59), although, in my best knowledge, there is no such evidence for medical doctors. So if You want to put doctors in "bad source of information" group, You need to reference it.

Limitations:

If You discuss the justification of supplements (like You did in lines 222-230) and if You will add some expansion of discussion section You should add limitation section. You need to clarify there that You actually do not test if supplement were probably used because i.e. You did not test if there are any insufficient intake in the diet.

Reviewer 2 Report

This study is very interesting and novel showing the supplementation in a sport little researched.

Comments:

Introduction

The introduction is adequate and perfectly structured

Page nº1 Lines 43-45

You indicate that according to reference nº 6 beach volleyball is played in these conditions. but there is no evidence of the world circuit being played in other temperature or humidity conditions? please indicate other humidity and/or temperature conditions

Page nº2 Lines 67-68

Only you mention two aspects that diminish sports performance, but I could point out some others? such as the nutritional factor or diseases derived from practicing sports

Methods

The methodology is complete and accurate

A minimal appreciation is made of the lack of a dietary evaluation of the athletes who complete the supplement questionnaire. Because one of the reasons for not taking supplements is that their needs are covered by food intake. Would there be any possibility to incorporate this information? Even if it is a short description this aspect would give more quality to the manuscript

Results

It would be necessary to indicate the relation of the intake of supplements in the category of “Dietary Supplements” (vitamins, calcium and iron) with respect to the RDAs.

Some of these supplements can be used to improve sports performance or treat an illness. For example, the use of iron can be used to prevent or treat iron deficiency with or without anemia. Could you point out some data in this regard?

Discussion

Were there any adverse effects related to supplementation? Could you discuss anything about this aspect?

I could discuss what differences and types of supplements are used between professional and recreational athletes

could you include a section on limitations of the study?

Reviewer 3 Report

Which are the nutritional supplements used by beach-volleyball athletes? A cross-sectional study at the Italian National Championship

********************

I found the study very interesting and necessary for public awareness. However, please consider the following:

Line 32: Try enriching the keywords. Use valid MeSH terms.

Line 33: Replace action by effort.

Line 49-50: A common belief? Robust clinical trials have assessed many nutrients, herbals and other types of dietary supplements on athletes' performance.

Line 52: Please update the references. Citations 8 and 9 are more than 15 years ago. There have been good position stands from ISSN and ACSM, and RCTs from different labs.

Line 53: Idem 52.

Line 53-56: Human physiology and the effects of dietary supplements on adaptation processes under certain phyiscal efforts are more important than the type of exercise itself. In this sense, the given example is obvious, and authors should center their analysis on molecular/cellular/physiological responses rather than the general point of view of "types of exercise".

Line 56-57: Wardenaar et al. 2017 (Nutritional Supplement Use by Dutch Elite and Sub-Elite Athletes: Does Receiving Dietary Counseling Make a Difference?) do not conclude that statement in their study. Please change for a valid citation regarding this sentence about consensus for the classification of nutritional supplements. Consider: Maughan et al. 2018 (COI), Kersick et al. 2018 (ISSN) or Thomas et al. 2016 (ACSM).

Line 72: replace "physical appearence" by "body composition"

Line 80: Authors do not report any detail of validation procedures (internal and external) neither fiability of the instrument. Only the Muwonge et al. 2017 is mentioned. Please provide a supplement file with the validated instrument.

Line 89: Do you have descriptive statistics of body mass, height and player's position?

Line 93: There was any kind of familiarization session or tutorial to fill the validated intrument?

Line 108: What type of carbohydrates did you refer to? Sacarose? Maltodextrin, High-molecular weight, Isomaltulose?

Also, which creatine and carnitine form? Or was in general? If so, why did not ask directly for information regarding the compounds with better outcomes in sports performance (i.e., creatine monohydrate and L-carnitine L-tartrate).

All above questions are important for the instrument validation.

Line 109-113: Did you run any pilot application of the instrument?

Line 119 - 121: citation missing.

Line 127: G2 likelihood-ratio

Line 139-140: Please correct. "...; AIC, SSABIC and G2 likelihood-ratio were lower in the 3-classes solution than in the 2-class one, whilst BIC was higher..."

Line 143: G2 likelihood-ratio

Line 155: carnosine phosphates? Did you mean "beta-alanine, phosphates"?

Line 165 - 166: Did you record more data from other variables (e.g., body mass, body composition, etc.)?

Line 175: Do you refer to professionals that hold a PhD title or to physicians?

Line 183: Replace the word "wheight" by "weight" in the Figure 2.

Line 187: is this a study, isn't?

Line 189: Keep the order and report the results according to your aims "... to evaluate the quantity and the heterogeneity of nutritional supplementation practices..."

Line 194: Check redaction.

Lline 197: I consider important to specify which products the authors refer. This not only would help to the reader but also (from a mechanistic / physiological point of view) would support some supplements.

Line 201-202: Which ones?

Line 203: Sports supplementation

Line 203-206: Human physiology and the effects of dietary supplements on adaptation processes under certain phyiscal efforts are more important than the type of exercise itself. In this sense, the authors should center their analysis on molecular/cellular/physiological responses and highlight that randomized clinical trials are needed in this population.

Consider also including recommendations reported in the ISSN position stands:

Protein by Jäger et al. 2017
Nutrient timing by Kerksick et al. 2017
Research & recommendations by Kerksick et al. 2018

Line 206: "To the authors' knowledge"? Or, would it be better "after a literature search on databases xxxxxx, xxxxx, xxxx, ..."

Line 207-208: Authors should be aware of the difference between "to investigate the efficacy of specific supplements on beach volley [assume that perfomance]" and "... explore the common practices".

Line 218: Delete the word "nutrient" before carbohydrates.

Line 221: What do you mean with "the entire set of amino acids"? Did you mean "essential amino acids" or "all amino acids?

Line 229-230: English editing.

Line 237: Do you refer to professionals that hold a PhD title or to physicians?

Line 244: It is important that authors highlight the fact of following sport nutrition guidelines (e.g., ISSN, ACSM, COI, etc.).

Line 249: "the authors"

Line 255: "fluid replacement"

Line 258: "aerobic ability"? Please do not use controversial terminology (see Chamari and Padulo, 2015).

Line 259: "fluid replacement recommendations"

Line 263: English editing

Line 267: More than "dehydration" is the "lack of fluid replacement" since reposition of ions is very important (e.g., avoid dilutional hyponatremia).

Line 270-273: English editing.

Line 263-276: It is important to conclude according to the aims of the study "to investigate the efficacy of specific supplements on beach volley [assume that perfomance]" and "... explore the common practices".

Check also:

- Limitations section is missing.
- Correct the word "dietician" throughout the manuscript.

Round 2

Reviewer 1 Report

The authors nicely respond to all my considerations. They add some important information in the method section and improved the discussion section with a good limitation paragraph. I am able to recommend this article for further processing.

Reviewer 2 Report

All the suggestions were resulted and incorporated into the manuscript. The study, in my opinion, is scientifically sound enough to be published. No further changes are necessary